# Orthogonal Drift Correction (ODC): Improving Semantic Alignment via Training-Free Embedding Refinement

## Abstract

Text-to-image models have achieved remarkable success in generating high-quality images from textual descriptions. However, they often struggle with "semantic drift," where the generated output fails to align perfectly with complex or nuanced text prompts. In this paper, we introduce Orthogonal Drift Correction (ODC), a novel, inference-time guidance technique designed to mitigate semantic drift without requiring any model retraining. ODC guides the image generation process through a two-stage mechanism. It first generates an initial image, then uses a pre-trained vision-language model to compute a semantic error vector between this image and the prompt. Next, we isolate the component of this error vector that is orthogonal to the prompt's direction, hypothesizing that this component represents the most detrimental, off-topic drift. By subtracting this orthogonal error vector, we create a refined conditioning vector for a second, corrected generation pass. Our experiments demonstrate that ODC significantly enhances prompt-image alignment, leading to images that more accurately reflect detailed compositional and attribute-based instructions. As a plug-and-play module, ODC offers a practical and computationally efficient method for improving the reliability of state-of-the-art text-to-image models.

## 1 Introduction

Modern text-to-image (T2I) models have opened new creative possibilities, allowing people to generate compelling images that would have been impossible to generate just a few years ago. Recent models such as Stable Diffusion (Rombach et al., 2022a), Imagen (Saharia et al., 2022), and DALL-E 2 (Ramesh et al., 2022) have demonstrated an unprecedented ability to synthesize visually compelling and artistic images from natural language descriptions, unlocking wide-ranging applications across numerous domains.

However, despite these profound capabilities, current state-of-the-art T2I models frequently fall short in generating images that precisely align with the full semantic meaning and composition of their input text prompts. This lack of precise semantic control, a phenomenon we term "semantic drift", pertains to the frequent failure of these models to precisely adhere to user prompts. For instance, they often misinterpret complex compositional instructions, leading to incorrect attribute binding (e.g., wrong material on objects), catastrophic neglect (Chefer et al., 2023) (omitting subjects from the prompt), or a failure to adhere to specified spatial relationships. This alignment gap forces users into frustrating trial-and-error prompt engineering, a practice that hinders the widespread and responsible adoption of these models.

This difficulty largely stems from how T2I models process textual information. State-of-the-art T2I diffusion models employ a U-Net architecture with cross-attention layers (Rombach et al., 2022a), which are crucial for fusing visual and textual features. While these layers contain rich semantic relations, the interaction can sometimes "diffuse" concepts. A variety of approaches have been proposed to address these limitations. Fine-tuning methods like DreamBooth (Ruiz et al., 2023) and Textual Inversion (Gal et al., 2022) personalize models for specific subjects but are computationally expensive and can suffer from language drift. Architectural changes like ControlNet (Zhang et al., 2023) and T2I-Adapter (Mou et al., 2024) offer explicit spatial control but require auxiliary inputs and specialized training. Another class of methods focuses on inference-time manipulation of attention maps or latent representations. Techniques such as Prompt-to-Prompt (Hertz et al., 2022), Null-text Inversion (Mokady et al., 2023), and Attend-and-Excite (Chefer et al., 2023) directly intervene in the generation process. While effective for specific tasks like editing or subject presence, they act as heuristics on the model's internal states and do not address the misalignment caused by imperfect initial text embeddings. This body of work reveals a clear gap for a method that can efficiently improve text-image alignment by directly correcting the text embedding to resolve complex compositional failures.

In this paper, we address this gap by introducing Orthogonal Drift Correction (ODC), a novel, inference-time technique that directly corrects the text conditioning vector to prevent semantic drift. Our approach is based on

the key insight that semantic drift originates from components in the initial text embedding that are orthogonal to the prompt's true semantic direction. ODC guides the image generation process through a two-stage mechanism. First, it generates an initial candidate image. Then, it leverages the powerful shared embedding space of a pre-trained vision-language model, to identify the semantic error vector between the generated image and the prompt. Crucially, instead of using the entire error, we isolate its orthogonal component via vector rejection. This component, which represents the "off-topic" deviation, is then subtracted from the initial text embedding. The resulting purified embedding guides a second generation pass, producing a final image with substantially higher prompt-image alignment.

Our contributions can be summarized as follows:

- We identify *orthogonal semantic drift* as a key source of prompt-image misalignment in text-to-image models, and propose a model-agnostic, training-free inference-time method that corrects the text conditioning embeddings without modifying the diffusion process or requiring fine-tuning.

- We introduce a vocabulary-based surrogate mechanism to approximate the orthogonal drift, re-embed it via the T2I model's text encoders, and refine it via an adaptive rank-reduced concept removal ($\mathcal{R}$), providing a principled and lightweight way to adjust conditioning representations.

- To support systematic evaluation, we contribute a dataset of 300 LLM-generated prompts specifically designed to probe semantic drift and prompt-image alignment.

## 2 BACKGROUND

This section provides the necessary context for understanding our proposed method, Orthogonal Drift Correction (ODC).

**Prompt-Image Alignment.** Prompt-image alignment refers to the faithfulness of a generated image to the full semantic meaning and compositional structure of its input text prompt. While the goal of Text-to-Image (T2I) models is to translate natural language into visually aligned results, achieving precise control over the spatial composition, complex layouts, poses, shapes, and forms described in a prompt remains a significant challenge (Huang et al., 2023; Saharia et al., 2022). This often necessitates numerous trial-and-error cycles from users to achieve their desired output.

**Text-to-Image Models.** The dominant architecture in modern T2I synthesis is the Latent Diffusion Model (LDM) (Rombach et al., 2022b), exemplified by systems like Stable Diffusion. Unlike earlier diffusion models that operated in the high-dimensional pixel space (Ho et al., 2020), LDMs perform the computationally intensive denoising process in the compressed latent space of a pre-trained variational autoencoder (VAE). This significantly reduces computational overhead while maintaining high-fidelity output.

The core of an LDM's denoising process is a U-Net architecture (Ronneberger et al., 2015), which comprises a series of downsampling and upsampling blocks with residual connections. Interspersed within this network are self-attention layers, which capture global spatial dependencies, and crucially, cross-attention layers. These cross-attention layers are the primary mechanism through which textual guidance is integrated into the image generation process. They allow the model to attend to different parts of the text prompt at each denoising step, conditioning the visual features being generated. More advanced models like Stable Diffusion XL (SDXL) (Podell et al., 2023) enhance this architecture with a larger U-Net and a dual text encoder system, combining the strengths of both CLIP (Radford et al., 2021) and T5 (Raffel et al., 2020) to achieve superior performance.

**Text Embeddings and Conditioning.** The journey from a text prompt to a guiding signal begins with tokenization, where the input string is converted into a sequence of discrete tokens. These tokens are then mapped to high-dimensional embedding vectors. This sequence of vectors is processed by a powerful text encoder—such as the CLIP text encoder or a larger language model like T5—to produce the final conditioning embedding.

This text embedding conditions the denoising U-Net at multiple resolutions via the cross-attention mechanism. At each cross-attention layer, the intermediate visual features (the query) attend to the text embedding (the key and value). This process generates spatial attention maps that define a rich semantic relationship between image regions and prompt tokens, critically influencing the final image's composition. Our method, ODC, intervenes at the very beginning of this pipeline. ODC plugs into the process by modifying the conditioning embedding *before* it is ever passed to the U-Net, thereby correcting the guidance signal at its source.

**Previous Attempts to Improve Alignment.** The challenge of prompt-image misalignment has spurred a wide variety of solutions. These can be broadly categorized into prompt engineering, training-based methods, and inference-time guidance.

- **Prompt Engineering:** At the most fundamental level, users engage in manual prompt engineering, iteratively refining text to achieve desired results. More structured approaches include prompt weighting, which allows users to amplify or attenuate the semantic influence of specific words, and the use of negative prompts. Negative prompts modify the unconditional embedding used in classifier-free guidance to explicitly steer the model away from unwanted concepts (Ho & Salimans, 2022). While useful, these techniques are manual heuristics. Our work can be seen as an automated and principled evolution, dynamically correcting the embedding based on model feedback rather than user guesswork.

- **Training-Based Methods:** A powerful but resource-intensive approach involves fine-tuning the model. Methods like DreamBooth (Ruiz et al., 2023) and Textual Inversion (Gal et al., 2022) specialize a model to a new user-provided concept by fine-tuning either parts of the model or a new "pseudo-word" embedding. Other techniques like LoRA (Hu et al., 2022) offer more parameter-efficient fine-tuning. While these methods excel at personalization and concept injection, they require significant computational resources for training and are not designed to solve general compositional failures for arbitrary prompts.

- **Inference-Time Guidance:** This category includes methods that, like ours, operate at inference time without retraining. A dominant technique is Classifier-Free Guidance (CFG) (Ho & Salimans, 2022), which amplifies the influence of the text prompt at each sampling step. While foundational, high CFG scales can degrade sample quality, a problem addressed by variants like Dynamic Thresholding (Saharia et al., 2022) and CFG++ (Chung et al., 2024). Another family of methods intervenes directly within the U-Net's sampling loop by manipulating cross-attention maps. Prompt-to-Prompt (Hertz et al., 2022) enables editing by injecting attention maps, while Attend-and-Excite (Chefer et al., 2023) directly optimizes attention values to mitigate catastrophic neglect.

Crucially, our work differs from existing inference-time approaches in its point of intervention. Whereas prior methods act by altering the internal dynamics of the U-Net denoising process, our approach is simpler and more direct: we refine the conditioning embeddings before diffusion begins. In contrast to training-based approaches, our method is entirely training-free, requiring no fine-tuning or additional optimization. Similarly, unlike attention-manipulation techniques, our intervention operates directly on the text embeddings, making it straightforward to implement and agnostic to the choice of U-Net architecture. By correcting semantic drift at its source, ODC provides a generalizable and principled way to correct semantic drift at its source, improving the alignment and reliability of pre-trained text-to-image models.

## 3 METHOD

Our proposed method, Orthogonal Drift Correction (ODC), is an inference-time guidance technique designed to enhance the semantic alignment of text-to-image models. The central hypothesis is that suboptimal prompt alignment, or *semantic drift*, is caused by components within the initial text embedding that are orthogonal to the prompt's primary semantic direction. ODC operates in a two-stage process. An initial image is generated and then evaluated against the prompt in a shared vision-language embedding space. The identified semantic error is then used to refine the initial text embedding for a second, more accurate generation pass.

### 3.1 PRELIMINARIES AND NOTATION

Let $p_{\text{txt}}$ denote the input text prompt. The text-to-image model consists of two text encoders, a token-level encoder $\mathcal{E}_{\text{tok}}$ and a pooled encoder $\mathcal{E}_{\text{pool}}$, together with an image generator $\mathcal{G}$ (typically a U-Net operating within a diffusion process). The standard generation process first encodes the text prompt as

$$\mathbf{E}_p = \mathcal{E}_{\text{tok}}(p_{\text{txt}}), \qquad \mathbf{e}_p = \mathcal{E}_{\text{pool}}(p_{\text{txt}}), \tag{1}$$

where $\mathbf{E}_p \in \mathbb{R}^{L \times H}$ is the sequence of per-token embeddings and $\mathbf{e}_p \in \mathbb{R}^H$ is a pooled sentence-level vector. The image generator then produces an initial image $I$ conditioned on both representations:

$$I = \mathcal{G}(\mathbf{E}_p, \mathbf{e}_p). \tag{2}$$

To evaluate and correct for semantic drift, we employ a pre-trained vision–language model (VLM), denoted $\mathcal{E}_{\text{VLM}}$, which embeds both text and images into a shared multimodal space.

### 3.2 THE ORTHOGONAL DRIFT CORRECTION (ODC) ALGORITHM

The ODC algorithm consists of two main stages, comprising six sequential steps in total. These steps are described in detail below and summarized in Algorithm 1. An overview of the process is illustrated in Figure 1.

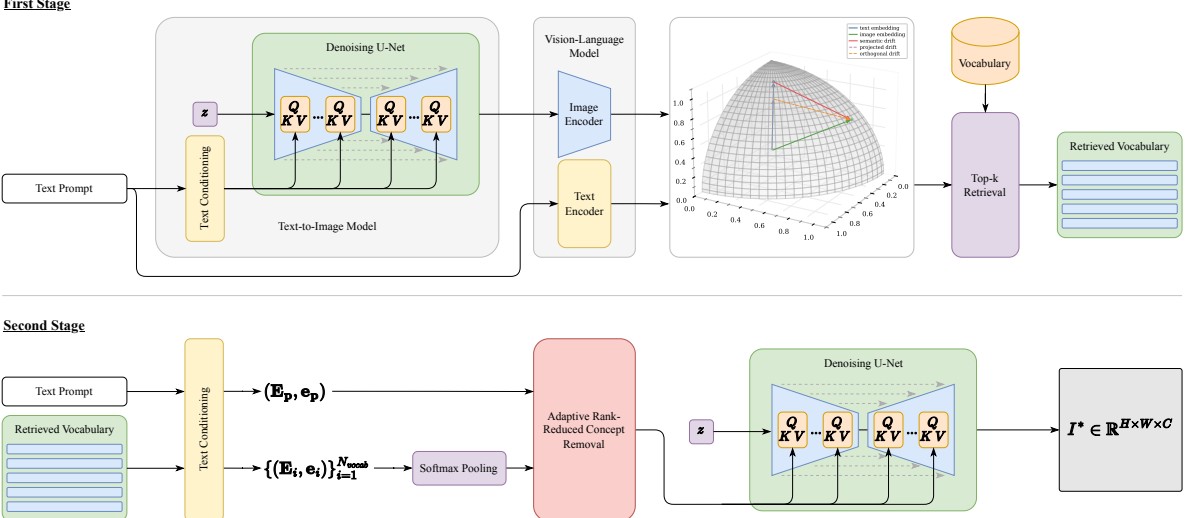

Figure 1: The workflow of Orthogonal Drift Correction (ODC). In the first stage, vocabulary items that best capture the orthogonal drift vector are retrieved. In the second stage, these vocabulary items are used in a concept removal operation, producing embeddings that better align with the user's text prompt and serve as input for the final image generation.

**Step 1: Initial Image Generation.** First, an initial image $I$ is generated using the standard process. The text prompt $p_{\text{txt}}$ is encoded by two complementary text encoders: one produces a sequence of per-token embeddings $\mathbf{E}_p$, while the other yields a pooled sentence-level representation $\mathbf{e}_p$. Both conditioning signals are then provided to the image generator $\mathcal{G}$ to synthesize the initial image.

$$\mathbf{E}_p = \mathcal{E}_{\text{tok}}(p_{\text{txt}}), \tag{3}$$
$$\mathbf{e}_p = \mathcal{E}_{\text{pool}}(p_{\text{txt}}), \tag{4}$$
$$I = \mathcal{G}(\mathbf{E}_p, \mathbf{e}_p). \tag{5}$$

**Step 2: Semantic Error Vector Calculation.** Next, we quantify the semantic discrepancy between the generated image $I$ and the original prompt $p_{\text{txt}}$. We use the $\mathcal{E}_{\text{VLM}}$ to project both into the shared embedding space. The *semantic drift vector*, $\mathbf{v}_{\text{drift}}$, is defined as the difference between these two embeddings. This vector represents the semantic error—the direction and magnitude of the deviation of the image's content from the prompt's intent as perceived by the VLM.

$$\mathbf{v}_{\text{text}} = \mathcal{E}_{\text{VLM}}(p_{\text{txt}}), \tag{6}$$
$$\mathbf{v}_{\text{img}} = \mathcal{E}_{\text{VLM}}(I), \tag{7}$$
$$\mathbf{v}_{\text{drift}} = \mathbf{v}_{\text{img}} - \mathbf{v}_{\text{text}}. \tag{8}$$

**Step 3: Orthogonal Drift Component Isolation and Vocabulary Retrieval.** The $\mathbf{v}_{\text{drift}}$ vector captures the total semantic error. We hypothesize that the most detrimental error components are those that introduce concepts extraneous to the prompt, which correspond to the direction orthogonal to the prompt's own embedding, $\mathbf{v}_{\text{text}}$. To isolate this component, we compute the vector rejection of $\mathbf{v}_{\text{drift}}$ from $\mathbf{v}_{\text{text}}$. This is achieved by subtracting the projection of $\mathbf{v}_{\text{drift}}$ onto $\mathbf{v}_{\text{text}}$ from $\mathbf{v}_{\text{drift}}$ itself. The projection, $\mathbf{v}_{\text{proj}}$, is calculated as:

$$\mathbf{v}_{\text{proj}} = \left( \frac{\mathbf{v}_{\text{drift}} \cdot \mathbf{v}_{\text{text}}}{\|\mathbf{v}_{\text{text}}\|^2} \right) \mathbf{v}_{\text{text}}. \tag{9}$$

The *orthogonal drift vector*, $\mathbf{v}_{\text{orth}}$, is then the rejection:

$$\mathbf{v}_{\text{orth}} = \mathbf{v}_{\text{drift}} - \mathbf{v}_{\text{proj}}. \tag{10}$$

To obtain a discrete, textual surrogate for $\mathbf{v}_{\text{orth}}$, we retrieve candidate terms from a custom vocabulary (detailed in Appendix A). Each vocabulary item $w$ is associated with an embedding $\mathbf{e}_w$, and cosine similarities are computed as:

$$s(w) = \frac{\mathbf{v}_{\text{orth}} \cdot \mathbf{e}_w}{\|\mathbf{v}_{\text{orth}}\| \, \|\mathbf{e}_w\|}. \tag{11}$$

The top-$k$ terms with the highest similarity scores are selected, forming a discrete approximation of $\mathbf{v}_{\text{orth}}$. We denote this retrieved set of tokens as $\mathcal{S}_{\text{orth}}$.

**Step 4: Approximating Orthogonal Drift in the Text Encoder Space.** These retrieved items are then re-embedded via the text encoders of the text-to-image model, thereby mapping the orthogonal drift signal back into the conditioning space used during generation.

These retrieved items $\mathcal{S}_{\text{orth}}$ are then re-embedded via the text encoders of the text-to-image model, yielding a per-token embedding matrix $\mathbf{E}_i$ and a pooled embedding vector $\mathbf{e}_i$ for each item $w_i \in \mathcal{S}_{\text{orth}}$. Since $\mathcal{S}_{\text{orth}}$ typically contains more than one item, we combine them using a softmax-weighted pooling to obtain equivalent representative embeddings:

$$\mathbf{E}_{\text{orth}} = \sum_i \beta_i \, \mathbf{E}_i, \tag{12}$$

$$\mathbf{e}_{\text{orth}} = \sum_i \beta_i \, \mathbf{e}_i, \tag{13}$$

where the weights $\beta_i$ are defined by the normalized cosine similarity scores

$$\beta_i = \frac{\exp(s(w_i))}{\sum_j \exp(s(w_j))}. \tag{14}$$

This results in a pair $(\mathbf{E}_{\text{orth}}, \mathbf{e}_{\text{orth}})$ that represents the surrogate set in the embedding space of the text encoders of our T2I model. These representative embeddings, in addition to the original embeddings, are supplied to the concept removal module.

**Step 5: Adaptive Rank-Reduced Concept Removal.** To remove undesired semantic concepts from the text embeddings, we employ an adaptive rank-reduction approach that automatically identifies and removes the meaningful semantic components while filtering out noise. We denote this operator by $\mathcal{R}$, and the procedure is summarized in Algorithm 2.

Given embedding tensors $\mathbf{E}_1, \mathbf{E}_2 \in \mathbb{R}^{B \times L \times H}$ (batch size $B$, sequence length $L$, hidden dimension $H$), we first determine the semantic rank of $\mathbf{E}_2$ through QR decomposition. For each batch element:

$$\mathbf{E}_2^T = \mathbf{Q}\mathbf{R}, \tag{15}$$

where $\mathbf{Q}$ contains orthonormal basis vectors. We identify semantically meaningful components by selecting basis vectors whose relative magnitude exceeds 1% of the primary component:

$$k = \max\left\{i : \frac{|r_{ii}|}{|r_{11}|} > 0.01\right\}, \tag{16}$$

where $r_{ii}$ are the diagonal elements of $\mathbf{R}$. This threshold empirically separates semantic information (typically ranks 3–15) from noise, as we observed that while $> 99\%$ of energy concentrates in the first component, the actual semantic distinctions distribute across multiple smaller components.

We then reconstruct a denoised version using only the top-$k$ basis vectors:

$$\mathbf{E}_2^{\text{reduced}} = (\mathbf{E}_2 \mathbf{Q}_{:,:k})\mathbf{Q}_{:,:k}^T. \tag{17}$$

The concept removal operates on flattened representations to capture global semantic relationships. Let $\tilde{\mathbf{E}}_1 = \text{flatten}(\mathbf{E}_1)$ and $\tilde{\mathbf{E}}_2 = \text{flatten}(\mathbf{E}_2^{\text{reduced}})$. We compute:

$$\tilde{\mathbf{E}}_{\text{out}} = \tilde{\mathbf{E}}_1 - \alpha \, \text{Rej}_{\tilde{\mathbf{E}}_1}(\tilde{\mathbf{E}}_2) = \tilde{\mathbf{E}}_1 \cdot \left(1 + \alpha \cdot \frac{\langle \tilde{\mathbf{E}}_1, \tilde{\mathbf{E}}_2 \rangle}{\|\tilde{\mathbf{E}}_1\|_2^2}\right) - \alpha \cdot \tilde{\mathbf{E}}_2, \tag{18}$$

where $\alpha$ controls the removal strength. The output is reshaped to the original dimensions. This adaptive approach automatically adjusts to concept complexity—simple concepts require fewer ranks while complex concepts utilize more—achieving robust removal without over-filtering.

**Step 6: Final Image Generation.** Finally, a new image $I^*$ is generated using the refined conditioning embeddings $(\mathbf{E}_{\text{refined}}, \mathbf{e}_{\text{refined}})$ produced by the concept removal module. The image generator $\mathcal{G}$ then synthesizes:

$$I^* = \mathcal{G}(\mathbf{E}_{\text{refined}}, \mathbf{e}_{\text{refined}}). \tag{19}$$

By incorporating the correction derived from the surrogate embeddings, the refined conditioning better suppresses drift components while preserving the original semantic intent of the prompt.

**Algorithm 1** Orthogonal Drift Correction (ODC)

1: **Input:** Prompt $p_{\text{txt}}$, vocabulary $\mathcal{V}$
2: **Output:** Final image $I^*$
3: $(\mathbf{E}_p, \mathbf{e}_p) \leftarrow (\mathcal{E}_{\text{tok}}(p_{\text{txt}}), \mathcal{E}_{\text{pool}}(p_{\text{txt}}))$
4: $I \leftarrow \mathcal{G}(\mathbf{E}_p, \mathbf{e}_p)$
5: $\mathbf{v}_{\text{txt}} \leftarrow \mathcal{E}_{\text{VLM}}(p_{\text{txt}})$
6: $\mathbf{v}_{\text{img}} \leftarrow \mathcal{E}_{\text{VLM}}(I)$
7: $\mathbf{v}_{\text{drift}} \leftarrow \mathbf{v}_{\text{img}} - \mathbf{v}_{\text{txt}}$
8: $\mathbf{v}_{\text{proj}} \leftarrow \frac{\mathbf{v}_{\text{drift}} \cdot \mathbf{v}_{\text{txt}}}{\|\mathbf{v}_{\text{txt}}\|^2} \mathbf{v}_{\text{txt}}$
9: $\mathbf{v}_{\text{orth}} \leftarrow \mathbf{v}_{\text{drift}} - \mathbf{v}_{\text{proj}}$
10: **for** each $w \in \mathcal{V}$ **do**
11: $\quad \mathbf{e}_w \leftarrow \text{embedding}(w)$
12: $\quad s(w) \leftarrow \frac{\mathbf{v}_{\text{orth}} \cdot \mathbf{e}_w}{\|\mathbf{v}_{\text{orth}}\| \|\mathbf{e}_w\|}$
13: **end for**
14: $\mathcal{S}_{\text{orth}} \leftarrow$ Top-$k$ items by $s(w)$
15: Re-embed $\mathcal{S}_{\text{orth}}$: $\{(\mathbf{E}_i, \mathbf{e}_i)\}_{w_i \in \mathcal{S}_{\text{orth}}}$
16: Compute weights $\alpha_i \leftarrow \frac{\exp(s(w_i))}{\sum_j \exp(s(w_j))}$
17: $\mathbf{E}_{\text{orth}} \leftarrow \sum_i \alpha_i \mathbf{E}_i, \quad \mathbf{e}_{\text{orth}} \leftarrow \sum_i \alpha_i \mathbf{e}_i$
18: $(\mathbf{E}_{\text{refined}}, \mathbf{e}_{\text{refined}}) \leftarrow \mathcal{R}\big((\mathbf{E}_p, \mathbf{e}_p), (\mathbf{E}_{\text{orth}}, \mathbf{e}_{\text{orth}})\big)$
19: $I^* \leftarrow \mathcal{G}(\mathbf{E}_{\text{refined}}, \mathbf{e}_{\text{refined}})$
20: **return** $I^*$

**Algorithm 2** Adaptive Rank-Reduced Concept Removal $(\mathcal{R})$

**Require:** $\mathbf{E}_1, \mathbf{E}_2 \in \mathbb{R}^{B \times L \times H}, \alpha \in [0,1], \tau = 0.01$; bounds $k_{\min} = 3, k_{\max} = 15$
**Ensure:** $\mathbf{E}_{\text{out}} \in \mathbb{R}^{B \times L \times H}$
1: **for** $b = 1$ to $B$ **do**
2: $\quad (\mathbf{Q}^{(b)}, \mathbf{R}^{(b)}) \leftarrow \text{QR\_decomposition}(\mathbf{E}_2^{(b)\top})$
3: $\quad \mathbf{r} \leftarrow \text{diag}(|\mathbf{R}^{(b)}|)$
4: $\quad k \leftarrow \max\{i : r_i/r_1 > \tau\}$
5: $\quad k \leftarrow \text{clip}(k, k_{\min}, k_{\max})$
6: $\quad$ Let $\mathbf{Q}_k^{(b)}$ be the first $k$ columns of $\mathbf{Q}^{(b)}$
7: $\quad \mathbf{E}_{2,\text{reduced}}^{(b)} \leftarrow (\mathbf{E}_2^{(b)} \mathbf{Q}_k^{(b)}) \mathbf{Q}_k^{(b)\top}$
8: $\quad \tilde{\mathbf{E}}_1 \leftarrow \text{flatten}(\mathbf{E}_1^{(b)})$
9: $\quad \tilde{\mathbf{E}}_2 \leftarrow \text{flatten}(\mathbf{E}_{2,\text{reduced}}^{(b)})$
10: $\quad \text{norm}_1 \leftarrow \|\tilde{\mathbf{E}}_1\|_2$
11: $\quad s \leftarrow 1 + \alpha \cdot \langle \tilde{\mathbf{E}}_1/\text{norm}_1, \tilde{\mathbf{E}}_2/\text{norm}_1 \rangle$
12: $\quad \tilde{\mathbf{E}}_{\text{out}} \leftarrow s \cdot \tilde{\mathbf{E}}_1 - \alpha \cdot \tilde{\mathbf{E}}_2$
13: $\quad \mathbf{E}_{\text{out}}^{(b)} \leftarrow \text{reshape}(\tilde{\mathbf{E}}_{\text{out}}, [L, H])$
14: **end for**
15: **return** $\mathbf{E}_{\text{out}}$

## 4 EXPERIMENTS

In this section, we introduce the experimental setup and the metrics used for our evaluation. Our goal is to evaluate whether our inference-time guidance technique improves prompt-image alignment across different models and challenging benchmarks, while preserving image quality and runtime efficiency.

### 4.1 EXPERIMENTAL SETUP

**Models.** To demonstrate the model-agnostic nature of ODC, we test it across different text-to-image models: Stable Diffusion XL (SDXL) (Podell et al., 2023) and FLUX.1 [schnell] (Black Forest Labs, 2024). For top-$k$ keyword extraction in the first stage, we use the BLIP2-ITM-ViT-G model (Li et al., 2023).

**Inference Parameters.** For our experiments, we generated images at a resolution of $1024 \times 1024$ pixels. For the SDXL model, we used the Euler Discrete sampler with 30 inference steps and a Classifier-Free Guidance (CFG) scale of 7.5. For the FLUX.1 [schnell] model, we employed the Flow Matching Euler Discrete sampler with 4 inference steps. The FLUX architecture does not utilize CFG. To ensure statistical robustness, we generate three images per prompt using fixed seeds across all methods and report the mean values for all metrics. While our method introduces several hyperparameters, we set them heuristically and did not perform an extensive search, in order to demonstrate that it functions as a robust inference-time guidance technique rather than one reliant on tuning. Specifically, we retrieved the top $k = 5$ vocabulary items, applied softmax pooling with temperature 0.5, and set the concept removal weighting parameter $\alpha$ to 1.0 for SDXL and 0.5 for FLUX.1 [schnell].

**Benchmarks and Prompts.** We evaluate all methods on the following datasets:

- **PartiPrompts** (Yu et al., 2022): A broad collection of prompts designed to probe imaginative and open-ended generation, highlighting generalization across everyday and creative scenarios. For our experiments, we filter the dataset to retain only prompts containing ten words or more, ensuring that all evaluated prompts provide sufficient semantic grounding.
- **AlignBench-300 (Custom Dataset)**: A curated collection of 300 prompts constructed by us to specifically test challenging alignment cases not emphasized in existing benchmarks. Details are provided in Appendix A.

**Baselines.** We compare ODC with baseline approaches to better understand its performance:

- **Vanilla**: The unmodified output from each base model, serving as our primary point of reference.
- **CFG-Sweep**: We report results from different CFG scales ($\{5.0, 7.5, 12.0\}$) to compare ODC against simply strengthening the prompt guidance.

- **Negative Prompt**: We create a tailored negative prompt for each generation by joining the retrieved vocabulary items, and we use it in place of the standard curated lists (e.g., 'bad anatomy, extra fingers, blurry').

## 4.2 EVALUATION METRICS

We employ a suite of automated metrics to assess text-image faithfulness and computational efficiency. To measure semantic alignment between a prompt and its generated image, we rely on both CLIPScore (ViT-L/14) (Hessel et al., 2021), a widely used metric for general semantic similarity, and BLIPScore (BLIP2-ITM-ViT-G) (Li et al., 2023), which leverages the more advanced BLIP-2 vision-language model for finer-grained evaluations. Given that our method operates during inference, we measure its computational overhead in terms of both latency—the average wall-clock time required to generate a single image—and GPU memory usage during generation.

## 5 RESULTS AND ANALYSIS

In this section, we present a comprehensive analysis of Orthogonal Drift Correction's performance. We begin with a qualitative evaluation to visually demonstrate the improvements in prompt-image alignment. We then present our main quantitative results across multiple benchmarks and models, followed by an ablation study to validate our core design choices. Finally, we discuss the limitations of our approach.

## 5.1 QUALITATIVE ANALYSIS

Visual inspection of the generated images provides the most intuitive evidence of ODC's effectiveness. In Figure 3, we present a side-by-side comparison of images generated by the vanilla Stable Diffusion XL model and our ODC-corrected method for a variety of challenging prompts.

The results clearly show ODC's ability to improve compositional accuracy and attribute binding, leading to reduced semantic drift. The baseline often struggles with correct modifier-object relationships and maintaining fine-grained attributes, while ODC generates images that align more accurately with the user description. Furthermore, for prompts that are vulnerable to spurious additions (e.g., "Victorian astronaut playing violin inside ornate greenhouse orbiting Saturn's rings"), the baseline model introduces unintended elements—such as additional violinists—that decrease alignment with the user's prompt. In contrast, ODC restricts generations to the requested content, preventing extraneous insertions and yielding outputs that more faithfully reflect the intended semantics. These qualitative examples suggest that by editing the initial text embedding, ODC provides a more reliable conditioning for the diffusion process. Additional qualitative results are provided in Appendix B.

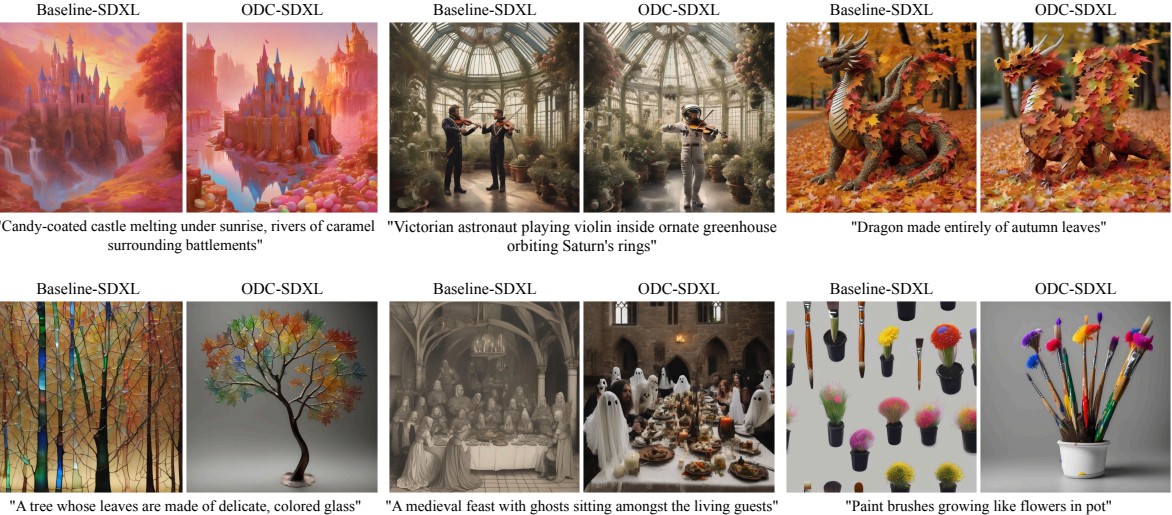

Figure 2: Qualitative comparison of images generated with and without Orthogonal Drift Correction (ODC). For each prompt, we show the output from the baseline SDXL model (left) and our ODC-corrected version (right).

## 5.2 Main Quantitative Results

To substantiate our qualitative findings, we present a thorough quantitative evaluation. As shown in Table 1, ODC consistently and significantly outperforms all baselines across all faithfulness metrics on both Stable Diffusion XL and FLUX.1 [schnell].

Across both datasets and for both model backbones (Stable Diffusion XL and FLUX.1 [schnell]), ODC consistently improves semantic alignment scores compared to existing baselines. For SDXL, our approach raises the CLIPScore by 1.438 and 1.667 points over the vanilla baseline on AlignBench-300 and PartiPrompts, respectively. On FLUX, we observe that our method can occasionally produce overcorrections, leading to a slightly lower CLIPScore on AlignBench-300 when taken directly. However, since the pipeline naturally produces both the original baseline image and the corrected variant, a simple best-of-two (BoT) strategy—returning whichever image achieves the higher score incurs no additional cost while reliably improving results. On FLUX, this strategy yields consistent gains over the baseline, and for SDXL it further amplifies the improvements already observed.

As shown in Table 2, ODC introduces some latency overhead while leaving memory consumption unchanged. Compared to the vanilla baseline, latency increases from 3.637 s/img to 7.56 s/img, reflecting the additional correction pass. Importantly, GPU memory usage remains constant at 23.015 GB, making the approach just as accessible in practice as the baseline methods. Overall, these results demonstrate that the gains in semantic alignment come with predictable and manageable efficiency costs.

Table 1: Main quantitative results on AlignBench-300 and PartiPrompts benchmark. We report alignment metrics across Stable Diffusion XL and FLUX.1 [schnell]. Best results are in **bold**.

| Method | AlignBench-300 | | | | PartiPrompts | | | |
| | Stable Diffusion XL | | FLUX.1 [schnell] | | Stable Diffusion XL | | FLUX.1 [schnell] | |
| | CLIP ↑ | BLIP ↑ | CLIP ↑ | BLIP ↑ | CLIP ↑ | BLIP ↑ | CLIP ↑ | BLIP ↑ |
|---|---|---|---|---|---|---|---|---|
| Vanilla | 30.307 | 0.159 | 29.297 | 0.159 | 30.833 | 0.152 | 31.182 | 0.152 |
| CFG-Sweep (CFG=5.0) | 29.953 | 0.159 | - | - | 30.448 | 0.151 | - | - |
| CFG-Sweep (CFG=12.0) | 30.396 | 0.160 | - | - | 31.125 | 0.152 | - | - |
| Negative Prompt | 28.937 | 0.157 | 28.297 | 0.158 | 30.083 | 0.151 | 30.568 | 0.152 |
| **ODC (Ours)** | 31.745 | 0.161 | 29.213 | 0.159 | 32.5 | 0.154 | 31.187 | 0.152 |
| **ODC-BoT (Ours)** | **32.205** | **0.162** | **30.292** | **0.161** | **32.99** | **0.154** | **32.21** | **0.153** |

Table 2: Efficiency analysis on SDXL. ODC introduces a fixed latency overhead from the initial generation pass and has no impact on GPU memory.

| Method | Latency (s/img) ↓ | Memory (GB) ↓ |
|---|---|---|
| Vanilla | 3.637 | 23.015 |
| ODC | 7.56 | 23.015 |

## 5.3 Ablation Studies

**The Importance of Orthogonal Correction.** Our central claim is that isolating and removing the *orthogonal* component of the semantic drift is key. To test this, we implemented a variant of our method, "Full Vector Correction," which uses the whole semantic drift vector ($\mathbf{v}_{\text{drift}}$) to retrieve the top-k vocabulary items. As shown in Table 3, while Full Vector Correction provides an improvement over the baseline, it underperforms our proposed ODC. This result strongly supports our hypothesis. Subtracting the full vector likely over-corrects by removing useful on-axis semantic information, whereas ODC surgically removes only the irrelevant semantic contents.

Table 3: Ablation results highlighting the importance of orthogonal correction on SDXL performance in AlignBench-300 (CLIPScore).

| Method | CLIP ↑ |
|---|---|
| Vanilla | 30.307 |
| Full Vector Correction | 31.489 |
| ODC | **31.745** |

## 5.4 LIMITATIONS

Despite its strong performance, ODC has several inherent limitations. First, its effectiveness is bound by the perceptual capabilities of the vision-language model (VLM) used in the first stage. If the VLM cannot see a specific semantic error, ODC cannot correct it. This is most apparent with highly complex and abstract concepts, which remain challenging for current VLM embeddings to represent accurately.

Second, the two-stage correction process inherently increases the inference latency compared to a standard single pass, as shown in Table 2. While the overhead from the embedding calculations themselves is negligible, the need for a full initial generation pass represents a direct trade-off between computational cost and semantic drift correction.

Finally, ODC is a guidance method, not a knowledge injection method. It can only refine the expression of concepts already understood by the base model. If a model has a fundamental knowledge gap (e.g., it does not know what a "gondola" is), ODC cannot inject this knowledge. Its role is to ensure the concepts the model *does* know are composed according to the prompt.

## 6 CONCLUSION

In this paper, we addressed the problem of semantic drift in text-to-image models by introducing Orthogonal Drift Correction (ODC), a novel, training-free, inference-time technique that guides the image generation through a two-stage process. Our extensive experiments demonstrate that ODC significantly enhances prompt-image alignment across multiple models and benchmarks. Without requiring any retraining or architectural changes, ODC provides a practical and generalizable solution for improving the alignment and reliability of existing pre-trained text-to-image models, making them more powerful creative tools.

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

## A  VOCABULARY DATASET AND ALIGNBENCH-300

### A.1  CONSTRUCTION OF THE VOCABULARY DATASET

To support our experiments, we constructed a large-scale vocabulary dataset designed to cover both visually grounded concepts and general linguistic phrases. The process involved several stages of aggregation, normalization, and filtering:

**Sources of candidate terms.** We drew upon three complementary sources:

- Open Images: object class labels provided in the official release.
- Visual Genome: object annotations, which supply a wide range of free-form object names contributed by annotators.
- Wikipedia: text spans extracted from English Wikipedia, from which we derived common noun phrases.

In addition, we used a curated English word list (the "web2" dictionary via the 'english-words' python package) to ensure broad lexical coverage.

**Normalization.**   All candidate strings were lowercased, Unicode-normalized, stripped of non-alphanumeric symbols, and reduced to tokens separated by single spaces. This step removed morphological noise such as capitalization, punctuation, and diacritics.

**Heuristic filtering.**   To eliminate obvious noise, we applied a series of rules including:

- length constraints (3-30 characters; 1-3 tokens),
- exclusion of strings dominated by digits or punctuation,
- removal of stopword phrases (e.g., "and", "the"),
- whitelist handling for short but useful terms (e.g., "tv", "pc").

**Spelling correction.**   Because crowdsourced annotations introduce idiosyncratic variants, we employed the *Sym-Spell* algorithm (edit distance 2) to canonicalize misspellings. This reduced duplication from typographic errors.

**Expansion with Wikipedia noun phrases.**   To augment the visual vocabulary with more general linguistic coverage, we parsed 1,000 Wikipedia articles using *SpaCy*'s dependency parser. From each article, we extracted noun chunks, retaining those appearing at least three times across the corpus. This yielded a complementary set of multi-word expressions such as "machine learning" or "climate change."

**Final merging and filtering.**   We combined the vision datasets (Open Images and Visual Genome), the curated dictionary, and the Wikipedia-derived phrases. A stricter regex-based filter (letters, apostrophes, and hyphens only) was applied to remove residual noise. This multi-stage process produced a broad and diverse vocabulary containing both concrete, visually grounded object names and higher-level conceptual phrases. In total, the final vocabulary contains $\approx 323000$ (after deduplication), which serves as the lexical backbone for the experiments reported in the paper.

## A.2   CONSTRUCTION OF THE ALIGNBENCH-300

In order to evaluate the semantic alignment between text-to-image models and user intent, we created a benchmark dataset of imaginative prompts, AlignBench-300. The goal was to establish a standardized set of inputs characterized by richness and diversity such that alignment failures would be clearly observable.

**Prompt Generation.**   We queried three distinct large language models—OpenAI o3 (o3-2025-04-16), Anthropic Claude Opus-4 (claude-opus-4-20250514), and Google Gemini 2.5 Pro (gemini-2.5-pro)—with instructions to propose creative and highly visual text-to-image prompts. Each model produced a large pool of candidate prompts emphasizing diverse styles, scenarios, and semantic complexity.

**Curation and the Resulting Dataset.**   From the generated pool, we manually reviewed and curated a final set of 300 prompts. This final collection of 300 curated prompts provides a compact but challenging benchmark that allows us to systematically probe how well text-to-image models respect compositional semantics. Unlike generic caption corpora, these prompts emphasize stress-testing of alignment through imaginative and specific requests.

## B ADDITIONAL QUALITATIVE RESULTS

In this section, we provide further qualitative results to complement the main paper. These examples illustrate our method's ability to improve prompt–image alignment, reduce semantic drift, and preserve visual fidelity across a diverse range of prompts.

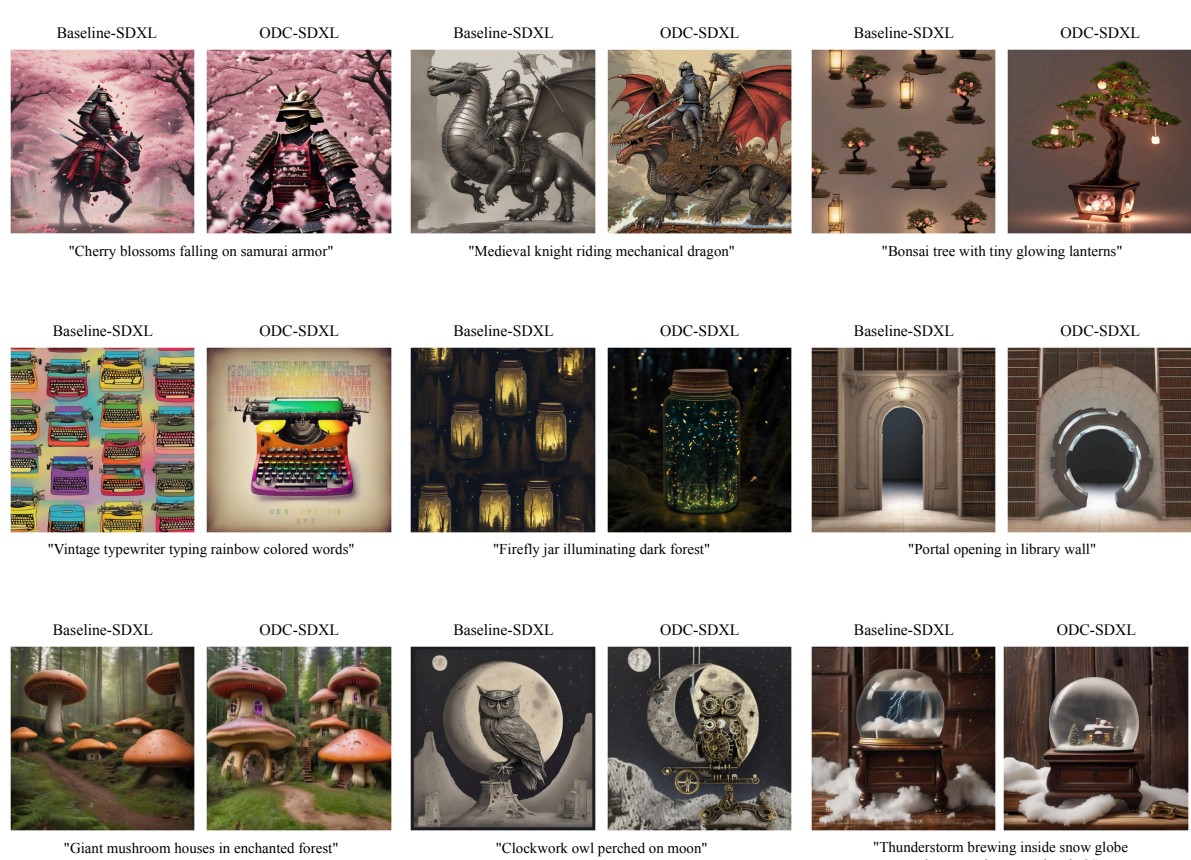

Figure 3: Additional qualitative results comparing baseline generations with Orthogonal Drift Correction (ODC). For each prompt, we show the output from the baseline SDXL model (left) and our ODC-corrected version (right).

