# OpenReview forum: "Orthogonal Drift Correction (ODC): Improving Semantic Alignment via Training-Free Embedding Refinement"
_ICLR.cc/2026/Conference — ICLR 2026 Conference Withdrawn Submission_

### Official Review · Reviewer_qYRt · 2025-10-25

**Soundness:** 1
**Presentation:** 2
**Contribution:** 3
**Rating:** 2
**Confidence:** 4

**Summary:**

The authors address semantic misalignment in text-to-image (T2I) models by guiding the generation process with a "semantic error vector." This vector is derived from an initial inference pass where the generated image and original prompt are fed into a Vision-Language Model (VLM). The method then isolates the component of this error vector that is orthogonal to the prompt's direction, using it to correct the subsequent generation.

**Strengths:**

The authors tackle the critical problem of image-text misalignment. They introduce a novel and interesting approach that leverages the semantic knowledge of a VLM to identify an error in the embedding space. Using this error representation to improve prompt alignment is a promising research direction.

**Weaknesses:**

The method's validation is insufficient. The only baselines used are a negative prompt, standard Classifier-Free Guidance (CFG), and vanilla generation. A significant omission, especially since the method incorporates a VLM, is a simple baseline of using the VLM to suggest a better prompt to fix the error.

Furthermore, a whole class of solutions for this problem is missing from both the related work and the baseline comparisons. This includes attention-based methods designed to fix semantic errors, such as Attend-and-Excite [1], and layout-conditioning methods like RAG-Diffusion [2].

Moreover, the evaluation relies on CLIP and BLIP scores, which are widely known to perform poorly on such nuanced tasks. These metrics often fail to detect subtle (and even overt) misalignments in attribute binding, spatial relationships, and object composition.

Finally, the reported performance gains are marginal. For instance, the CLIP score on Vanilla FLUX only increases from 29.297 to 30.292 using the proposed method. I suspect this slim margin is not statistically significant.

[1] Attend-and-Excite: Attention-Based Semantic Guidance for Text-to-Image Diffusion Models (Chefer et al. 2023)

[2] Region-Aware Text-to-Image Generation via Hard Binding and Soft Refinement (Chen et al. 2024)

**Questions:**

1. What is the final size of the Parti dataset after you apply your filter?

2. Why did you filter for prompts longer than 10 tokens? This filtering choice is not justified.

3. There are no qualitative results for the FLUX model shown, making it impossible to visually assess the method's impact on this state-of-the-art generator. Why were these omitted?

---

### Official Review · Reviewer_UB8o · 2025-10-27

**Soundness:** 2
**Presentation:** 2
**Contribution:** 2
**Rating:** 4
**Confidence:** 4

**Summary:**

This paper proposes Orthogonal Drift Correction (ODC), a novel training-free method to correct semantic drift in text-to-image (T2I) models. ODC identifies and removes the orthogonal component of semantic error in the text embedding space using a two-stage process: it first generates an image, computes the semantic drift via a Vision-Language Model (VLM), and then refines the text embedding by removing the orthogonal ”off-topic” component before regenerating the image. Experiments on SDXL and FLUX.1 show improved prompt-image alignment without fine-tuning or architectural changes.

**Strengths:**

Novel and Intuitive Approach: The idea of decomposing and removing the orthogonal semantic drift is innovative and well-motivated.

Training-Free and Model-Agnostic: ODC works with pretrained models without fine-tuning, making it easily applicable.

**Weaknesses:**

Dependence on VLM Capability: The correction quality is limited by the VLM’s ability to detect semantic errors.

Increase Inference Time: The two-stage generation doubles latency, which may hinder real-time use.

Cannot Inject New Knowledge: ODC only rearranges or suppresses existing concepts in the model; it cannot teach the model new ones.

**Questions:**

1.We all know that the generation process has a degree of randomness and is influenced by the random seed. Does a single image have sufficient representativeness in the VLM's embedding space?

2.If the Initial Image Generation is only used to determine the V_{text} in the embedding space, can a different model be used for it compared to the Final Image Generation?

---

### Official Review · Reviewer_qk2P · 2025-10-29

**Soundness:** 3
**Presentation:** 2
**Contribution:** 3
**Rating:** 4
**Confidence:** 4

**Summary:**

The authors address the problem of semantic drift in text-to-image diffusion models, where the generated image fails to semantically align with the input prompt. They propose a plug-and-play, training-free method called Orthogonal Drift Correction (ODC). The key idea is to perform two-stage generation to remove components in the initial text embedding that are orthogonal to the true semantic direction of the prompt, thereby enhancing text adherence in existing T2I models.

**Strengths:**

1. The paper is generally well-structured and complete.
2. The proposed method of removing orthogonal components from the input text embeddings provides a insightful perspective on mitigating semantic drift.

**Weaknesses:**

1. The authors assume that semantic drift originates from the initial text embeddings, whereas some prior works[1] attribute it to the initial latent noise. The paper does not provide an explanation or experimental analysis to justify this assumption. In addition, the proposed approach relies on a VLM for identifying orthogonal components, which could itself introduce additional bias, and the method’s effectiveness may depend heavily on the capability of the chosen VLM.
2. The experiments are not sufficiently comprehensive. Since the task focuses on semantic alignment in T2I generation, the authors should include comparisons with stronger baselines[2], such as CONFORM[3], Attend-and-Excite[4], and other recent methods that explicitly address alignment. Moreover, semantic alignment is a broad concept that also includes attribute binding, counting, and spatial relation tasks. The paper would benefit from additional benchmarks[4,5] covering these aspects to better demonstrate the effectiveness of the proposed method. Ablation studies are also limited — since the method depends on a VLM, experiments using different VLMs would help analyze robustness and dependency.
3. The proposed method requires two generations, which increases inference time, but the authors do not provide any discussion or analysis regarding this computational cost.
4. Although experiments are conducted on Flux Schnell, there is no qualitative visualization or analysis for this model. In the quantitative results, the improvement over the vanilla baseline appears relatively minor, raising questions about the practical effectiveness of the method.
5. In terms of writing, the method section and workflow image are not clearly presented, and several equations seem redundant or overly detailed.
6. It would be valuable for the authors to show examples of the retrieved vocabulary obtained from the orthogonal projection for each prompt, as this could help clarify the model’s behavior and provide qualitative insight into what the correction actually removes.

[1]: InitNO: Boosting Text-to-Image Diffusion Models via Initial Noise Optimization

[2]: Token Merging for Training-Free Semantic Binding in Text-to-Image Synthesis

[3]: CONFORM: Contrast is All You Need For High-Fidelity Text-to-Image Diffusion Models

[4]: Attend-and-Excite: Attention-Based Semantic Guidance for Text-to-Image Diffusion Models

[5]: T2I-CompBench: A Comprehensive Benchmark for Open-world Compositional Text-to-image Generation

**Questions:**

Is there any experimental evidence supporting the hypothesis that semantic drift originates from the initial text embeddings?
Did the authors perform any intermediate analyses, such as examining the retrieved vocabulary items corresponding to the orthogonal direction?
Additional related questions are discussed in the Weaknesses section.

---

### Official Review · Reviewer_n34s · 2025-11-01

**Soundness:** 2
**Presentation:** 2
**Contribution:** 2
**Rating:** 4
**Confidence:** 4

**Summary:**

The generated image could be misaligned with the text prompt semantically therefore this paper introduced Orthogonal Drift Correction (ODC), an training-free method that first using vision-language model to compute the semantic error vector and then further isolate the semantic error vector, so the original prompt could be modified based on this to improve the performance.

**Strengths:**

1. The idea is simple and easy to implement.
2. They presented some examples where the performance is improved on the semantic level.
3. The method is clearly illustrated with both algorithm and section text.

**Weaknesses:**

1. The writing needs improvement. For example, Step 1 in the method is just regular image generation process, which could be merged with other parts. Figure 1 has two stages but the method part never mention that but in 4 steps.

2. Results presentation are inadequate. For example, only 6 pairs are presented in the main paper and 9 more examples. The author should show more categorized results like in Human, animal, animation, etc.

3.  The same goes to ablation study, where no qualitative examples are presented.

4. No human evaluation is presented.

**Questions:**

1. After improved semantic alignment, does it also help other aspects such as aesthetic score?

2. Does it work on other architecture or even non-diffusion model or diffusion transformer model?

3. "we observed that while > 99% of energy concentrates in the first component, the actual semantic distinctions distribute across multiple smaller components." Could you clarify this with statistic number?

---

### Note · Authors · 2025-12-03

I have read and agree with the venue's withdrawal policy on behalf of myself and my co-authors.